# Russia’s National Concept to Reduce Alcohol Abuse and Alcohol-Dependence in the Population 2010–2020: Which Policy Targets Have Been Achieved?

**DOI:** 10.3390/ijerph17218270

**Published:** 2020-11-09

**Authors:** Maria Neufeld, Anna Bunova, Boris Gornyi, Carina Ferreira-Borges, Anna Gerber, Daria Khaltourina, Elena Yurasova, Jürgen Rehm

**Affiliations:** 1Institute for Clinical Psychology and Psychotherapy, TU Dresden, Chemnitzer Str. 46, 01187 Dresden, Germany; jtrehm@gmail.com; 2WHO European Office for Prevention and Control of Noncommunicable Diseases, Moscow, Leontyevsky Pereulok 9, 125009 Moscow, Russia; ferreiraborgesc@who.int; 3Institute for Mental Health Policy Research, Centre for Addiction and Mental Health (CAMH), 33Ursula Franklin, Toronto, ON M5S 2S1, Canada; 4National Medical Research Center for Therapy and Preventive Medicine of the Ministry of Health of the Russian Federation, Petroverigskiy Pereulok 10, 101990 Moscow, Russia; asbunova@gmail.com (A.B.); BGornyy@gnicpm.ru (B.G.); 5Sechenov First Moscow State Medical University (Sechenov University), 109004 Moscow, Russia; gerber.anna.p@gmail.com; 6Federal Research Institute for Health Organization and Informatics of Ministry of Health of the Russian Federation, Dobrolyubov Street 11, 127254 Moscow, Russia; khaltourina@gmail.com; 7WHO Office in the Russian Federation, Leontyevsky Pereulok 9, 125009 Moscow, Russia; yurasovae@who.int; 8Campbell Family Mental Health Research Institute, CAMH, 250 College Street, Toronto, ON M5T 1R8, Canada; 9Institute of Medical Science (IMS), University of Toronto, Medical Sciences Building, 1 King’s College Circle, Room 2374, Toronto, ON M5S 1A8, Canada; 10Department of Psychiatry, University of Toronto, 250 College Street, 8th Floor, Toronto, ON M5T 1R8, Canada; 11Dalla Lana School of Public Health, University of Toronto, 155 College Street, 6th Floor, Toronto, ON M5T 3M7, Canada

**Keywords:** alcohol policy, alcohol affordability, alcohol mortality, alcohol use disorders, health behaviors, lifestyle, life expectancy, noncommunicable diseases, prevention, Russia

## Abstract

In the 2000s, Russia was globally one of the top 5 countries with the highest levels of alcohol per capita consumption and prevailing risky patterns of drinking, i.e., high intake per occasion, high proportion of people drinking to intoxication, and high frequency of situations where alcohol is consumed and tolerated. In 2009, in response to these challenges, the Russian government formed the Federal Service for Alcohol Market Regulation and published a national strategy concept to reduce alcohol abuse and alcohol-dependence at the population level for the period 2010–2020. The objectives of the present contribution are to analyze the evidence base of the core components of the concept and to provide a comprehensive evaluation framework of measures implemented (process evaluation) and the achievement of the formulated targets (effect evaluation). Most of the concept’s measures were found to be evidence-based and aligned with eight out of 10 areas of the World Health Organization (WHO) policy portfolio. Out of the 14 tasks, 7 were rated as achieved, and 7 as partly achieved. Ten years after the concept’s adoption, alcohol consumption seems to have declined by about a third and alcohol is conceptualized as a broad risk factor for the population’s health in Russia.

## 1. Introduction

Although national [1,2,3,4] as well as international studies [5,6,7,8,9] constantly drew attention to the exceptionally high levels of alcohol consumption and alcohol-attributable harm in the Russian Federation in the 1990s and the early 2000s, national regulations covering alcohol were essentially focused on halting the rise of illegally produced alcoholic beverages and regaining control over the alcohol market [10,11]. The Soviet legacy of a paternalistic stance towards the individual’s health choices has resulted in passive attitudes towards individual responsibility of lifestyle choices and help-seeking behaviors, especially in relation to heavy alcohol use and smoking, high-fat diet, and insufficient health-promoting exercise [12,13,14]. This, along with the government’s heavy dependence on alcohol sales as an important source of tax revenue, has resulted in high mortality rates, especially in middle-age working-class males, and a considerable gap in life expectancy between Russia and Western countries [10,15,16].

However, beginning in 2000 with the formation of the state-owned distillery enterprise “Rosspirtprom”, various actions were taken to regulate and restructure the alcohol market. Rosspirtprom took more than half of Russian spirits producers under formal control, which made the government the biggest spirits producer in the country, with established annual production quotas and a recently introduced minimum unit price on vodka [10,17]. In his 2005 annual message to the Federal Assembly of the Russian Federation, the President officially acknowledged the problem of hazardous alcohol use and alcohol-dependence at the population level, and emphasized the significant number of alcohol poisoning deaths in young men and the role of surrogate alcohol in the unfortunate statistic of 40,000 fatal alcohol poisonings per year [18]. In the same year, substantial amendments to the main alcohol law of the Russian Federation were introduced, the Federal Law of 22 November 1995 No. 171 “On State Regulation of the Production and Turnover of Ethyl Alcohol and Alcoholic Products” [19]. The new provisions mainly addressed unrecorded alcohol, i.e., alcohol that is not recorded and taxed as beverage alcohol because it is produced or sold outside of the formal channels under governmental control, according to the common definition of the World Health Organization (WHO) [20]. A more effective denaturing procedure of non-beverage alcohol was imposed, along with substantial tax increases on non-beverage alcohol, the introduction of new anti-counterfeit excise stamps, and stricter licensing provisions for alcohol producers, such as new provisions of a large prepaid capital, which, subsequently, eliminated a lot of smaller distilleries that were believed to produce large shares of alcohol that they did not declare to authorities in order to avoid paying taxes [10,21]. Also, a centralized monitoring system for collecting data on produced volumes of alcohol, including the use of raw materials and leftovers, was introduced. The so called EGAIS system (which, in English, stands for Unified State Automated Information System) was made mandatory for every producer of ethanol and spirits, and enabled additional surveillance and control over a large segment of the alcohol market, thus eliminating further sources of unrecorded alcohol [22,23]. Starting from 2006, regional authorities were given the right to restrict hours of alcohol sales, but not many regions used this opportunity to restrict availability of retailed alcohol [24].

By the end of 2008, and following a presidential decree, the Federal Service for Alcohol Market Regulations (“Rosalkoregulirovanie”) was established as a federal executive body under the Prime Minister and a mandate for developing and implementing regulations concerning production and distribution of ethyl alcohol and alcoholic products was handed over to this new agency, including licensing of the entire distillery branch [25,26].

Still, all the measures listed above were aimed at the alcohol market, and additional federal policies targeted the population, individual drinking behaviors and health choices, were not introduced until 2009, when the Civic Chamber of the Russian Federation published an analytical report on alcohol consumption and alcohol-related harm in Russia, providing a set of recommendations on counter-actions [27]. In the same year, following consultations held by the Chairman of the Government of the Russian Federation and the President [28], the “Concept of state policy to reduce alcohol abuse and prevent alcoholism among the population of the Russian Federation for the period until 2020” was published in the form of an order of the Government of the Russian Federation and signed by the President [29]. The concept was written as a strategy paper with goals, priorities, implementation mechanisms, and suggested measures to reduce alcohol consumption and alcohol use disorders (AUDs) at the population level, including measures involving the newly formed Federal Service for Alcohol Market Regulations. The concept also featured a set of concrete results aimed to be achieved until 2020.

Several studies document a steep decline in alcohol consumption and alcohol-attributable harm as well as all-cause mortality in Russia in the past decade, with most of them linking this, in one way or another, to alcohol control policy [10,11,21,22,30,31,32,33,34].

Although some evaluations of the 2009 policy concept were carried out, often as part of the overall changes of the Russian legislations towards promoting healthy lifestyles and increasing life expectancy [35,36,37,38,39], none of these analyses followed a formal evaluation process with a predefined framework and formalized rating procedures. The present article therefore presents a more formalized multi-step evaluation approach. It provides an overview of the different components of the concept and offers an evaluation framework for those measures that were or are currently being implemented (process evaluation), and which targets have been achieved, wherever this is possible (effect evaluation). The aim of this formal process is not only to assess to what extent the concept is based on evidence-based policy options [40,41] and whether the set indicators were achieved, but also to identify possible gaps and areas in the current regulations, and where there is still room for improvement.

The analysis also aims at identifying areas where alcohol regulations could be improved more than 10 years after the adoption of the 2009 concept, and where further analysis and evaluation is required to assess impact in the future, also in view of the next strategy concept to be slated in 2020.

## 2. Materials and Methods

A complementary mixed-method framework of evaluation was chosen to assess the level of implementation of the different measures suggested in the 2009 concept as well as the achievement of the indicators outlined.

In the first step, a directed content analysis [42,43] of the concept document was carried out by the first three authors. An independent coding was performed of the different core components of the concept and their relations to each other. A separate coding was then performed for the policy measures and the key indicators of the concept as well as their relationship to an evidence-based policy portfolio of the 10 areas of action developed by the World Health Organization (WHO) as part of the Global Strategy to Reduce Harmful Use of Alcohol [44] and the European Action Plan to Reduce the Harmful Use of Alcohol 2012–2020 [45]. Each measure of the concept was then assigned to an area of the WHO portfolio, creating a formal evaluation framework.

In the second step, the concept’s measures were independently rated on whether they were generally evaluable as based on the information provided by the concept itself and on the available data sources within the national context. Methods and data sources for evaluation were then discussed and agreed upon: see the Appendix A for a more detailed description of sources.

In the final step, a mixed-methods evaluation of the measures implemented (process evaluation) and the achievement of the formulated indicators (effect evaluation) was carried out for the period 2008–2018. For the latter, a simple rating on an ordinal 3-point scale was carried out to highlight if indicators were achieved, partly achieved, or not achieved. Differences in the initial coding and final ratings were thoroughly discussed among all authors until a consensus was reached.

## 3. Results

### 3.1. Overview of the Different Components of the Concept Paper

The main areas and the core components of the concept are presented in Table A1 of the Appendix A. The concept document has a brief introductory section on the general provisions and the current situation on alcohol abuse in the Russian Federation, followed by specific sections on objectives, guiding principles, priority tasks, information support, implementation measures and mechanisms, main stages of implementation, and anticipated results as well as defined indicators to assess the level of progress and to evaluate the results.

The main indicators of achieving implementation results are defined as the following, divided into two separate stages:First stage (2010–2012):
-A 15% reduction of alcohol per capita consumption.-Reduction of the share of spirits in total alcohol consumption alongside a significant decrease of the total level of alcohol consumption.-Increasing the level of involvement of children and youth in sports.Second stage (2013–2020):
-Liquidation of the illegal alcohol market.-A 55% reduction of alcohol per capita consumption, as well as the creation of conditions for further downward trends in alcoholic consumption.-Reduction of the incidence of morbidity and mortality from alcohol-dependence, including alcoholic psychosis.

In the following, we focus on the evaluation of the specific measures described in the concept paper and the related areas for action of the WHO alcohol policy portfolio and provide a rating on whether they can be viewed as implemented based on the available information (process evaluation). The concept contained a total of 21 measures, which were assigned to 8 out of the 10 action areas of the WHO portfolio, and most were rated as having been implemented. For an overview of this rating, please see Table 1. The developments in different areas are briefly described to highlight the most important changes.

Components were rated as evaluable within the framework of evaluation if any kind of information (e.g., legislative acts, research articles, websites, etc.) could be identified by the raters, which indicated that any kind of change had taken place since 2009 in relation to the specific measures (e.g., whether alcohol media campaigns had been carried out or not). Evaluation methods and data sources were discussed and suggested for each of the measures prior to the formal analysis, which was carried out per the assigned WHO area of policy action (see below). Further sources were identified and added during the evaluation process.

### 3.2. Leadership, Awareness, and Commitment and Community and Workplace Action

Both the WHO Global Strategy to Reduce the Harmful Use of Alcohol [44] and the European Action Plan to Reduce the Harmful Use of Alcohol 2012–2020 [45] recognize “leadership, awareness, and commitment” by national and local governments as a recommended target area of action and as an important cornerstone of alcohol policy in a given country. Following the 2009 concept strategy, various legislative changes occurred in this action area, marking a turning point in Russian health policy towards lifestyle issues and individual health behaviors. The Federal Law No. 171-FZ “On state regulation of the production and turnover of ethyl alcohol and alcohol products”, which is the main instrument of alcohol control policy, has been amended several times since then, in line with the 2009 concept paper. Some of the most important amendments were likely the 2011 amendments (some of which came into force in 2012), when the legislation was renamed in federal law in “On state regulation of the production and circulation of ethyl alcohol, alcoholic, and alcohol-containing products, and on limiting the consumption (drinking) of alcoholic products”, which expanded the law’s scope to regulate on- and off-premise availability of alcoholic beverages as well as their consumption in public places, and also included beer in the scope of this legislation [46]. Before this, beer was regulated through special decree and provisions, and beginning in 2005 by a special federal law on beer regulation [47], so including it into the federal law helped with harmonizing the existing alcohol provisions [11].

A series of amendments were made to the Tax Code, the Administrative Code, the Criminal Code, as well as various federal laws of the Russian Federation, aimed at tightening alcohol control.

Moreover, a series of new laws were introduced beginning in 2009, aimed at promoting a healthy lifestyle for the population with a special focus on reducing alcohol consumption. For instance, in 2019, the Ministry of Health launched a series of corporate health promotion programs as part of their “Strengthening the Worker’s Health Approach”, that was aimed not only at the prevention of occupational diseases and injuries, but also at promoting healthy behaviors among workers [48]. One of the specific modules featured in the attachment of the document was the “Reducing alcohol consumption with harmful consequences” module, which suggested routine workplace alcohol testing to decrease the risk of occupational injuries and related sick leave rates as well as routine screening for individual risk levels with the Alcohol Use Disorders Identification Test (AUDIT) and providing consultations to workers at risk. This, however, would require training of specialists (doctors, paramedical workers, nurses, or psychologists) as well as facilities where screening and counseling could be carried out. If occupational psychologists were already present in the company, they were advised to take up the function of screening and providing consultations to the workers at risk within their offices. Moreover, the document suggested developing corporate alcohol awareness communication campaigns and making corporate events alcohol-free.

Increasing healthy life expectancy as well as the share of individuals committed to a healthy lifestyle were set as key targets of the National Projects of the Russian Federation in 2018. For instance, the National Projects “Demography” and “Healthcare” encompassed various different sectors and there was a big overlap between the WHO areas of community and workplace actions and the health system’s response (for more details on the changes to the latter, see below). The recently adopted 2020 “Strategy for the formation of a healthy lifestyle for the population, prevention and control of non-communicable diseases for the period up to 2025” has two main goals: (1) the formation of a healthy lifestyle for the population, and (2) the prevention and control of non-communicable diseases. In relation to alcohol, the strategy sets a clear target: the reduction of per capita alcohol consumption to 9 L of pure alcohol by 2025 [49]. Also, the 2020 changes to the Russian constitution expanded the notion of “coordination of healthcare issues” in Article 72G by adding “affordable and high-quality medical care, preserving and strengthening public health, creating conditions for a healthy lifestyle, forming a culture of responsible citizens’ attitudes to their health” as fundamentals of the constitutional system [50].

After the 2009 national strategy on alcohol was released, changes in different areas followed, spanning from very concrete regulations on alcohol pricing, marketing, etc., that are more thoroughly discussed below, to general awareness campaigns as well as overall health promotion efforts.

Following an order of the Ministry of Health in 2009, a network of health centers was established as within the state-run polyclinics, the main providers of primary healthcare in Russia, the main aim of which was to promote a healthy lifestyle among citizens of the Russian Federation, with a special focus on reducing alcohol and tobacco use [51]. As a result, people of all ages can now undergo a thorough health exam free of charge once a year that includes height and weight measurement, electrocardiography, blood sugar and cholesterol level tests, and an evaluation of the respiratory system functions. Patients also receive individual counseling on their individual health behaviors and risk factors. Moreover, various centers organize so-called “health schools”—a series of educational seminars on behavioral risk factors and programs on tobacco cessation, physical activity, and prevention of diabetes and hypertension [52]. Various other decrees and strategies have been released since then, dedicated to the development of a healthy lifestyle in the population, also accounting for regional needs (for an overview, see Table A2 in the Appendix A).

Also beginning in 2009, a series of media campaigns was launched by the government in order to raise awareness for the harmful use of alcohol and tobacco in the population and to promote the newly formed health centers and their early check-ups for risk factors and potential health issues (see Table A3 of the Appendix A). It is worth noting that while the earlier campaigns from 2009 to 2011 mainly dealt with severe clinical outcomes such as alcohol-dependence and alcoholic psychosis, the recent media campaigns were more focused on alcohol as a broad risk factor for health in general and promoted an alcohol-free lifestyle, along with healthy nutrition and regular physical exercise.

Special attention was given to the promotion of a healthy lifestyle among youth, often at a regional level and involving youth volunteer programs, in various campaigns [53,54].

For instance, the Ministry for Sports, Tourism, and the Ministry of Health used the 2018 FIFA World Cup as an opportunity to promote ideas of health and physical activity among young Russians, although the government made concessions to FIFA and temporarily loosened some of the national marketing restrictions on beer [11].

Overall, physical activity has been increasing in Russian adolescents, while alcohol use has been declining [55,56], and cohort analyses suggest that the documented decline in Russian drinking was mostly driven by younger people [57,58,59].

### 3.3. Pricing Policies and Sales of Alcoholic Beverages

Pricing is an area with the largest evidence base globally in its effectiveness and impact on consumption levels and mortality outcomes. It has been repeatedly demonstrated that individuals’ alcohol consumption is largely influenced by the affordability of alcoholic beverages, which in turn is determined by prices of alcoholic beverages in relation to other products, as well as by consumers’ incomes and rates of inflation [60]. There are various ways to decrease alcohol affordability, but alcohol excise taxation has been recognized as the most cost-effective policy to prevent alcohol-related harms and, along with reducing alcohol availability and marketing, as one of the WHO’s “best buys” of alcohol control [41,61]. 

From 2008 onwards, excise rates on alcoholic beverages were gradually raised and will be raised as per the Russian Tax Code gradually until 2022. However, when adjusted for inflation via the annual consumer price index (CPI) percentage changes for alcoholic beverages, the real tax increase happened only for the period 2009–2014, followed by a decrease and a period of relative stagnation with moderate increases projected for the next three years (Figure 1). The decision not to raise excise rates on alcoholic beverages for the period 2014–2016 and 2018–2019 resulted in a de-facto decrease of duty rates during these periods (for more details on inflation-adjusted and non-adjusted rates: Table A4a,b of the Appendix A). The inflation-adjusted rates are expected to rise again only starting from 2020 onwards, but will be de facto lower than at the level of 2014, when accounting for inflation. 

The overall excise increases for the period 2008–2022 are higher for lighter alcoholic beverages (with an ABV with 9% and below) than for alcoholic beverages with a higher ethanol concentration than 9% ABV, covering vodka and spirits. For lighter alcoholic beverages, the inflation-adjusted excise rate increase for the period 2018–2022 is 41%, and it is 12% for stronger alcoholic beverages. The largest excise increases were observed for beer, with 189% increase for usual beer (i.e. beer with 0.5–8.6% ABV) and 66% increase for strong beer (beer with 8.6% ABV and above) and more fluctuations were observed in the levels of inflation-adjusted beer excise rates over time (see Figure 1, for more details: Table A4a,b of the Appendix A). 

In 2016, separate excise rate categories were introduced for domestically produced wines, sparkling wines, as well as for imported ones, presumably supporting the domestic winemaking culture as a distinct measure of the national concept. Overall, the increases were the steepest between 2010 and 2014.

Another way of decreasing the affordability of alcohol is to introduce minimum prices below which alcoholic beverages cannot be sold. This is especially important for cheap alcoholic beverages, typically preferred by heavy drinkers. In Russia, this is also considered as a counter-measure to reduce unrecorded alcohol use as any alcoholic beverages that are sold below a certain limit can readily be identified as counterfeits [62,63,64].

Although a minimum retail price on vodka and other spirits with an alcohol content of >28% was introduced in 2000, this measure remained largely unnoticed because the price was too low in relation to the per capita income of the population. Attempts to fix and control alcohol prices per governmental decrees and to introduce a minimum price on vodka go back to the early 1990s [10,25].

Following the concept’s strategy, minimum retail prices for vodka and spirits with >28% ABV were gradually raised from 2010 onwards, following a long-term strategy of alcohol price increases announced by the Federal Service for Alcohol Market Regulation [65] (for an overview in changes in minimum prices, see Figure 2). In 2011, a minimum retail price at a much higher level was introduced for cognac and brandy, which are defined separately as per Federal Law No. 171, which regulates alcohol. Moreover, minimum wholesale prices for raw ethanol were introduced in 2014, at a different level for ethanol produced from food and non-food raw materials (see Appendix A
Table A5).

In 2015, the minimum price for vodka was actually decreased following an intervention from the President, who voiced concerns that high alcohol prices might lead to an increase in unrecorded consumption [66]. For the period 2014–2016, excise rates on alcohol were no longer increased (see Appendix A
Table A6), and their pace of increase has slowed down considerably for the remaining period. However, a minimum retail price for sparkling wine was introduced in 2015 and, starting in 2016, minimum prices and excise rates were raised again. These measures had an impact on the affordability of alcoholic beverages, as can be seen in Figure 3.

Here, we define affordability as a simple measure, namely how many liters of pure alcohol of a given beverage could be bought with an average Russian per capita income in a given year. As shown in Figure 3, compared to other alcoholic beverages, vodka was always the cheapest and most affordable alcohol in any given year, although its affordability has changed the most over time.

Until 2010, affordability of all alcoholic beverages (with the exception of beer) was increasing due to both an increasing population income and excise rates lagging behind inflation. The raising of the minimum retail price of vodka in 2010 substantially decreased its affordability, along with the substantial increase in excise rates. Beginning in 2015, with the decrease in the minimum price and the freezing of excise rate on the level of 2014 for two subsequent years, vodka became more affordable again. Affordability of fortified wine and beer has been generally decreasing since 2010 against a backdrop of increasing excise taxes, while affordability of wine and sparkling wine has remained more or less at the same level, with a general increase until 2013 and a slight decline since then. Despite the introduction of a minimum price on cognac in 2011, its affordability increased between 2008 and 2013, slightly decreased until 2016, and has been rising since then. Out of all analyzed beverages, cognac remains the least affordable alcohol.

The steepest affordability drop was observed for vodka, as both higher excise rates and minimum prices have pushed up its price. However, the decrease of the minimum price and the freezing of the excise rates in 2015 led to an immediate rebound effect and affordability increased again. For the last period, 2016–2018, affordability of all alcoholic beverages has either stagnated or increased again, although at a much slower pace, which reflects the more moderate raise in excise rates and minimum prices as compared to the 2010–2014 period.

When looking at the recorded sales of alcoholic beverages for the period 2008–2018, a more or less steady decline is observed for all beverage types with the exception of cognacs, as sales have slightly increased by 6% for the latter (see Figure 4). Wine products decreased by 7%, beer and beer-based products by 30%, and sparkling wines by 35%. The largest sales decrease was observed for beer and spirits, by 53%. However, all sales have slightly increased for 2018. Overall, recorded sales of all alcoholic products declined by 33% for the 2008–2018 period. The same level of decrease was reported for total per capita consumption (including the use of unrecorded alcohol) for the same time window [11]. It should be noted, however, that the per capita consumption refers to a rate and takes into account population change, whereas the provided sales figures are not controlled for by population size.

Against the backdrop of these changes, a substantial shift in the beverage-specific structure in total alcohol sales occurred, with beer surpassing vodka as the most frequently sold beverage (see Figure A1 of the Appendix A). In 2016, the Federal Service for Market Regulation, which was responsible for implementing these long-term pricing strategies to ensure the flow of tax revenues of alcohol to the state, was subordinated to the Ministry of Finance, which subsequently transferred the mandate of developing and implementing alcohol control policies to itself [67].

### 3.4. Reduction of the Public Health Impact of Illicit Alcohol and Informally Produced Alcohol and Improving Monitoring Systems

One of the most frequently voiced arguments against alcohol price increases is the possibility of increased consumption of unrecorded alcohol. Indeed, price increases may lead to such a substitution effect in the absence of regulation, but the existing evidence suggests that a full substitution of recorded by unrecorded consumption could never be reached, or that it could be successfully prevented if specific measures addressing unrecorded alcohol are put in place [40].

Unrecorded alcohol consumption has been a known issue in Russia for decades, with research showing that specifically consumption of non-beverage alcohol and other surrogates are tightly linked to more dangerous drinking patterns, social deprivation, (often undiagnosed) AUDs, and greater risks of alcohol-attributable harms, including premature mortality [7,68,69,70,71], especially in working-age men.

However, the Russian government has been addressing unrecorded alcohol for years. Laws and key measures were introduced as early as 2005/2006 to decrease first and foremost the use of surrogate alcohol (for an overview, see Reference [66] and Appendix A
Table A6). One of these crucial measures was the introduction of the so-called Unified State Automated Information System (Russian abbreviation: EGAIS), which was developed in 2005 to state-register and monitor the volumes of produced beverage alcohol and spirits at production sites and the use of the leftovers [21,22]. Despite grave technical issues at the beginning, the system was upgraded and significantly improved over time. Starting from 2016, EGAIS was introduced stepwise for wholesale and retail sales of alcohol, including raw pharmaceutical alcohol in urban and rural settings, covering all alcoholic beverages, although some limitations do exist for beer, cider, and mead [72]. Producers and distributors had to buy EGAIS equipment and register within the system, which pushed many of the smaller companies and retailers suspected of selling illegally produced and/or untaxed alcoholic beverages out of the market [10,21].

The implementation of EGAIS into retail sale made illegal sales much more difficult as it allowed monitoring of the entire supply chain, allowing the final consumer to track the purchased bottle from the production plant to the cash register at the point of sale using a QR code on their purchase receipt via a mobile application. For more details on the system, see References [11,73].

Today, EGAIS is one of the most advanced monitoring systems globally, relying on processing of data in real time, and although it does not cover the production and sales of cosmetic and medicinal alcoholic products, which are often misused as surrogates, its implementation was associated with decreasing mortality rates of alcohol-attributable as well as all-cause mortality [10,33].

Additional anti-counterfeit legislations followed in 2017, after a mass methanol poisoning with counterfeited methanol-based bath lotions occurred in 2016/2017 in Irkutsk [23]. Amendments to the Federal Law No. 171 as well as the Criminal Code of Russia were made, introducing harsh penalties for counterfeiting. A series of decrees of the Consumer Protection Service (Rospotrebnadzor) introduced temporary bans on the sales of antiseptics in 2017–2018, until a government decree adopted a permanent ban on the sale of non-beverage alcoholic products with an ABV of >28% at a lower price than the established minimum retail price for vodka and spirits, to discourage their misuse as surrogates [74].

Estimates of the World Health Organization indicate that for the period 2008–2016, the share of unrecorded alcohol among total alcohol consumption remained relatively stable at about 30% (see Figure 5) and that no substitution effect between recorded and unrecorded alcohol was observed for the general population as both indicators were decreasing over time [75,76]. 

### 3.5. Restricting Availability of Alcohol

Although regional authorities had the opportunity to introduce time restrictions in off-premises settings as early as 2006 due to the associated amendments of the Federal Law No. 171, not many regions limited temporal availability of alcohol until 2011, when the amendments to the federal law on alcohol introduced a nation-wide night ban on alcohol sales from 11 PM to 8 AM local time [11]. However, regional authorities were allowed to adopt longer limits beyond this, and it was documented that regions that introduced longer restrictions or implemented this measure earlier than others experienced a stronger decline in alcohol per capita alcohol consumption [78,79]. Overall, restricting evening hours of sale has shown to be more effective than limiting morning hours and no general substantial effect with unrecorded alcohol has been found, which is in line with the existing international literature in the field [40,80]. The regional distribution in off-premise hours of sale as of 2020 is presented in Figure 5 (more details can be found in Table A7 and Figure A2 of the Appendix A).

Similar to the findings on the effects of price increases on unrecorded consumption, studies have shown that the time restrictions on alcohol sales have not led to overall increases in unrecorded alcohol use, although this seems to vary across regions and sub-populations [24,78].

Moreover, the same series amendments to the Federal Law No. 171 of 2011–2012, considerably restricted places where alcohol sales where allowed, prohibiting alcohol sales at petrol stations, medical, educational, cultural and sport facilities, as well as within a certain perimeter around them, and introduced higher fines for facilities selling or serving alcohol to minors, which went along with better enforcement of the minimum drinking age regulation. The expansion of this legislation’s scope now covered beer as an alcoholic beverage and regulation of alcohol drinking as a behavior: a ban on alcohol consumption in public (including public transport) and places of mass gatherings was introduced with the exception of on-premises serving locations (for an overview, see Reference [11]). Still, the current legislation does not restrict density of outlets in any form, although there are discussions to have density regulations at a local level.

Sales of alcoholic beverages over the Internet were prohibited as early as 2007 and before the adoption of the national concept, but various Internet sellers of counterfeit alcoholic beverages could be found in recent years. For instance, in 2017, more than 25 online sellers of counterfeit alcohol were offering spirits at a price 6–15 times lower than the average retail sale prices and online purchases could be made without any form of age identification [68].

### 3.6. Restricting Marketing of Alcoholic Beverages

Although Russia has introduced strict advertising provisions for alcohol as early as the 1990s, marketing regulations were considerably expanded over time, also considering the emergence of new media and marketing formats. In line with the outlined changes in the areas of alcohol pricing and availability, important changes in alcohol marketing occurred in 2011, when amendments to the Federal Law No. 38 “On the Advertisement” were adopted, introducing important restrictions of content and advertising spaces [81]. The 2012 amendments to the same law prohibited the alcohol advertising on the Internet, including social networks [82]. In 2014, alcohol advertising was prohibited in sports and recreational facilities and at a distance of less than 100 m from the buildings that accommodate these facilities [83]. However, in the same year, the provisions of the federal law were loosened again, allowing advertisement of domestically produced wine and sparkling wine or wine made of domestic grapes on TV and radio between 11 PM and 7 AM, except for live broadcasting and broadcasts of children’s and youth sports competitions. Also, advertising of beer and beer-based beverages were allowed during live broadcasts or in recording of sports competitions (including sports matches, games, battles, races), except for children’s and youth sports competitions, as well as on sports TV and radio channels, following the concessions made to the FIFA committee in the context of the World Cup in Russia. However, the latter provisions were repealed in 2019, again prohibiting the placement and distribution of advertisements for beer and beer-based beverages during live broadcasts or in recordings of sporting events and during official sporting events [84].

However, some loopholes in the associated regulations need to be mentioned. Although there is a digital ban on alcohol advertising in place, regulated by both the federal law on advertising and the federal law on alcohol, there are still legal ways to promote alcohol online. For instance, website content of alcohol producers and distributors and their presence on social media is not recognized as alcohol advertising as long as they present their content as a means of sharing information about their products and prices, including special offers. The only provisions that must be followed are to include a filter question on age for website visitors before the website can be entered (to prevent people under 18 from browsing the website), to provide a health warning on the risks of “excessive use of alcohol” on the website, and to not encourage alcohol use in the displayed materials [85,86]. Although the legislation prohibits the indirect promotion of brands, such as in the form of advertising alcohol-free beer, it is not very well enforced. Also, sales pitches through faux analytical techniques are not prohibited, i.e., a situation where alcohol brands organize alleged opinion polls about their products but are using this opportunity as a hidden promotion technique rather than for collecting data.

Advertising and promotion of alcohol at the points of sales is also not restricted and sellers use creative ways to promote alcoholic beverages on-site, for instance through special “loyal client” programs and other pricing schemes and incentives, such as “buy two, get the third one free” special offers, sales, and lotteries [87,88]. Similarly, alcohol advertising is not prohibited in on-premises serving locations, where not only posters and banners depicting alcoholic beverages are allowed, but also the promotion of brands in the form of prints on furniture, sunshades, tableware, etc. [87].

### 3.7. Health Services’ Response and Decreasing Morbidity and Mortality Indicators

Traditionally, prevention of alcohol use disorders in Russia was centered on preventive measures of alcohol-dependence as within the specialized sub-discipline of psychiatry, known as narcology. Prevention, treatment, and rehabilitation of substance use disorders, including detoxification procedures, are carried out in the specialized system of narcology, as established by the protocols of the Ministry of Health, and government-run as well as private providers exist. Starting from 2011, narcological help has been provided in accordance with a national law; before that, it was regulated by various national and regional regulations [89]. Government-run narcological services are free of charge, but specific barriers to treatment exist as narcological registration means imposed monitoring procedures that limit employment opportunities and have other implications, e.g., a possible withdrawal of one’s driver’s license [11,90]. Although narcological services are required by law to offer anonymous treatment to a certain share of patients who “for various reasons avoid reaching out to narcological facilities”, this is limited to a certain contingent as based on local health system’s capacities [91]. Avoiding narcological registration and associated stigmatization as the main reasons why a large, yet unknown, proportion of individuals with AUDs turn to private narcological services as the private sector is heavily involved in providing narcological health services in Russia, with some studies suggesting that its share has been growing over time [38].

The narcological system does not offer any interventions for individuals with risky patterns of alcohol consumption that do not fulfill the clinical criteria for harmful use of alcohol and alcohol-dependence in accordance with the Russian abridged version of the 10th revision of the International Statistical Classification of Diseases and Related Health Problems (ICD-10), and these individuals do not consider narcology as an option due to the monitoring requirements and the associated stigma.

In order to provide preventive interventions for people who are at risk but do not meet any clinical criteria and also to amend narcological monitoring conditions, changes were made to the associated legislation that regulates the provision of medical help in Russia. In 2011, a new federal law called “On the basics of protecting the health of citizens in the Russian Federation” was adopted, which de facto forbids compulsory treatment and monitoring of individuals with AUD (or who fulfill the ICD-10 criteria of AUD as per established narcological diagnosis) and introduced informed consent protocols as the basis for any kind of narcological intervention. In 2014, a “Concept of Modernization of the Narcological Service of the Russian Federation” was issued as per Order of the Ministry of Health, which had a strong focus on prevention [92]. In the following years, several changes to the narcological provisions were made, introducing the notion of “primary specialized care”, namely the presence of narcology specialists in primary healthcare (PHC) facilities, to whom patients with risky alcohol use could be referred by other healthcare workers. Further changes were made in 2019, when narcological monitoring procedures were amended [93]. Narcological patients can no longer be monitored if the narcological services fail to provide an examination within one year or if the patient is imprisoned for a term of one year or above. Moreover, narcologists can decide to release the patient from monitoring within one year if the patient moves to another jurisdiction, presents a written refusal to be monitored, or in the case of the death of the patient. This 2019 change therefore offers a legal possibility to avoid narcological monitoring and its implications but, as for now, no data or materials exist that would document how this legislation is applied in the treatment setting (for an overview of changes in narcology provisions, see Table A8 of the Appendix A). Moreover, materials that address screening for alcohol use and potential AUDs and other alcohol-related harms were developed by narcology specialists at that time [94,95,96,97].

As for other areas of the health system and the occurred changes, the same 2011 Federal Law on healthcare laid the groundwork for reorganizing the preventive services and consolidating efforts to prevent noncommunicable diseases, reduce risk factors, and facilitate the promotion of a healthy lifestyle at the population level [98]. Aligned with this law, the Ministry of Health issued a decree in 2012 [99], which formulated a new framework for the carrying out of dispanserization procedures. These procedures involve preventive check-up activities undertaken at the population level as organized within PHC facilities. They go back to the Soviet centralized public health system and they are still in place in some form in Russia as well as some other post-Soviet countries.

Dispanserization is carried out for all individuals regularly, although different schedules for different population groups exist, and includes preventive and specialized medical examinations for early detection and prevention of diseases.

The 2013 legislative changes have also included specific evidence-based screening procedures for risk factors as part of a two-step screening process. The “risk of harmful alcohol use” was explicitly included in the document, and the supporting methodological guidelines for the various risk factor screening procedures have introduced the CAGE screening tool (the name of which is an acronym of its four questions) for the alcohol component [100,101,102].

Starting in 2013, dispanserization measures were aimed at detecting risky alcohol use at the level of PHC facilities and offering “risk factor correction”, which, however, remained a vague term since no further efforts were made to develop materials and train staff to address alcohol use within the PHC framework beyond the methodological guidelines of dispanserization,. Still, a series of other decrees on dispanserization have followed since then, each of them changing the procedures slightly [103]. Starting in 2018, the CAGE was replaced by the Alcohol Use Disorders Identification Test (AUDIT), which is now carried out as part of the two-step screening process of dispanserization, where its short version, AUDIT-C, is administered at first contact and the full test is done once a certain threshold (>3 points for women and >4 for men) has been reached. Experts have noted that the AUDIT is a better instrument in detecting risk levels in Russia and a thorough adaptation and validation procedure of this test has just been completed to further facilitate the implementation of screening and brief interventions in Russian PHC facilities [104,105,106]. Therefore, the current dispanserization guidelines instrumentally define “risk of harmful alcohol use” as an AUDIT score above a certain cut-off and code this as Z72.1, as per the ICD-10 [102].

Figure 6 highlights the age-standardized rate of the risk of harmful alcohol use for the period 2013–2019 for men and women, as based on available screening data of dispanserization for individuals of 21 years and older (other dispanserization measures for younger age groups apply and only data for this populating group was available). Rates for men were 3–4 times higher than for women and were steadily decreasing over time, with the steepest decline observed for 2013–2016. Rates for women slightly increased between 2013 and 2014, then substantially decreased until 2017 and have been rising since then. The gender gap has somewhat narrowed over the last years, but due to the methodological changes in the screening procedures as well as the non-representative nature of the data, it is difficult to interpret the trends.

Treatment data on the prevalence and incidence of alcohol-dependence (F10.2), alcoholic psychosis (F10.5), and harmful use of alcohol (F10.1) as registered within state-run narcology services have demonstrated a general decline over the last years [11]. For the period 2008–2018, the incidence of alcohol-dependence diagnosed in narcology clinics has declined by 56% (including alcoholic psychosis), the incidence of alcoholic psychosis by 69%, and the incidence of harmful use diagnoses by 70%, fully achieving the concept’s indicators (for more details, see Figure A3 in the Appendix A). Since narcological help can be offered anonymously by private healthcare and is thus outside of this monitoring system, these trends need to be interpreted with caution, especially since some authors suggest that the share of anonymous treatments has been growing over the years [38]. At the same time, alcoholic psychoses are unlikely to be treated by private healthcare providers routinely due to the acute state of the patients, which is why the observed decline seems to reflect the overall trend.

The same downward trend was also observed for mortality from alcohol-dependence and alcoholic psychoses as well as other causes of death that are 100% alcohol-attributable (see Figure 7 and References [11,33]).

However, mortality rates for alcohol poisonings, alcoholic cardiomyopathy, and alcoholic liver disease stagnated during the period of frozen excise rates and decreased minimum prices of vodka and spirits, followed by another decline.

### 3.8. Rating of the Priority Tasks

An overview of the concept’s priority tasks and their targets as well as the rating of their achievement as part of the effect evaluation is presented in Table 2. A simple rating on an ordinal 3-point scale was carried by the first three authors to highlight if indicators were achieved, partly achieved, or not achieved. Each priority task and the associated targets were rated separately, as based on the overall evaluation performed in each action area. Differences between the initial raters were thoroughly discussed among all authors and a consensus was reached.

Out of the 14 priority tasks, 7 were rated as having been achieved, and 7 as partly achieved.

Overall, the reduction of total alcohol per capita consumption and a relative reduction of spirits consumption has been achieved as a target, which has happened against the backdrop of introducing pricing and taxation measures to reduce affordability of alcoholic beverages and specific measures combating production and sale of unrecorded alcoholic products. Still, consumption of spirits is quite high as compared to the rest of the WHO European Region. The establishment of the Federal Service for Alcohol Market Regulation seems to have been a decisive step in increasing the efficiency of alcohol market regulations, specifically in relation to price control, while the introduction of various other laws has considerably tightened alcohol control as part of a broader legislative framework to promote health behaviors and a healthy lifestyle.

All-Russian opinion polls as well as WHO data demonstrate that abstention rates have been increasing over the last years, while overall awareness on the risks of alcohol have been rising, along with the population’s support of alcohol policies (for more details, see Table A9 of the Appendix A and References [11,107,108,109].

Promotion of a healthy lifestyle was a task carried out not only through the now reformed services for preventive medicine, but also as part of many other areas, including media, sports, education, culture, and tourism, including regional initiatives, the Russian volunteer movement, as well as the newly established Presidential Grants program [54,110,111,112,113].

## 4. Discussion

As already outlined in the specific sections, almost all of the measures and targets of the Russian Concept of state policy to reduce alcohol abuse and prevent alcoholism among the population of the Russian Federation for the period until 2020 were rated as achieved or as partly achieved. Some of the components could not be evaluated to their full extent because no specific evaluation indicators were identified and/or because the relevant data was inaccessible.

Almost all components were aligned with the WHO portfolio of evidence-based actions to reduce harmful use of alcohol and the only WHO action area the concept did not feature were drink-driving policies as well as measures to decrease the negative impact of alcohol such as health warnings and server training. However, compared to the rest of the WHO European Region, these policies are relatively well implemented in Russia, with the big exception of server training [11].

Before we proceed to discussing the implications of the findings for the next national strategy to come, some obvious limitations to this evaluation study must be mentioned. First of all, as all the analyses of effect evaluation carried out were descriptive, we have to assume that various other factors might have influenced the trends, especially morbidity and mortality trends as well as screening outcomes on the harmful use of alcohol as quality of assessment and coding varies regionally, especially in a country as large and as diverse as Russia. For instance, it has been documented that a certain proportion of alcohol poisoning deaths are misclassified as deaths due to cardiovascular diseases in Russia, and this misclassification seems to not be uniform across regions as coding practices vary, mainly depending on local capacities of forensic bureaus and hospitals providing regular autopsies [114,115,116]. Similar issues in recording alcohol-attributable causes of death as well as any other alcohol-related outcomes might occur in various regions, possibly with varying levels of stigma due to local cultural norms, which would bias the evaluation results in an unknown manner [117,118,119]. For instance, some national reports imply that screening data on alcohol as per dispanserization vary from region to region, also partly due to local data collection issues [120]. Therefore, the provided calculation of risk of harmful alcohol consumption in Figure 5 should be treated with caution, and no trend interpretation based on this calculation is possible because changes in screening procedures have occurred during the observation period.

Also, not only data quality, but also implementation and enforcement of alcohol policies are known to vary greatly across the Russian regions. Overall, Russia, as the world’s largest country, is very diverse: more than 180 ethnic groups live in the more than 80 regions, some of which are autonomous republics populated mostly by non-Russian ethnic groups. A considerable proportion of Russia’s population is Muslim, at almost 12%, and this proportion varies greatly across regions, shaping the cultural and societal norms towards alcohol use as well as the level of associated stigma. Therefore, regional analyses are needed to complement the outlined national trends.

Finally, it is likely that some measures or areas were overlooked as part of this rating and evaluation and that some measures were not accounted for in this analysis.

The overall evaluation results clearly demonstrate that Russia has introduced the three “best buys” as part of the 2009 concept and that great achievements have been made, specifically in the area of pricing policies (i.e., taxes and minimum prices) with the establishment of a federal service overseeing their implementation, although pricing policies were temporarily loosened in 2015. The establishment of the EGAIS system to ensure alcohol tax payments were being collected and reduce illegal sales of alcohol seems to have played an important role in the observed developments.

However, unrecorded alcohol remains one of the main challenges of alcohol control in Russia: despite a total decrease during the observed time window, its relative share remained stable at a level of about 30% of total consumption with a large proportion of surrogate alcohol among all unrecorded products consumed [11,121]. Despite a series of specific legislations, some important gaps remain as various types of unrecorded alcohol remain available in Russia, especially in the form of alcoholic beverages produced by legal manufacturers, but not declared to authorities to avoid taxation (e.g., the so-called “third shift” vodka), as well as non-beverage alcohols commonly misused as surrogates by heavy drinkers and people with AUDs [23,64,70,122,123]. Since alcohol-based medicinal compounds as well as certain cosmetic alcohols are not subject to the Federal Law No. 171 and associated taxation provisions, they are cheaper than alcoholic beverages and are not tracked via the EGAIS-monitoring channels. Currently, the Tax Code of the Russian Federation exempts medicinal compounds as well as alcohol-containing perfumery and cosmetic products “in small containers” from its list of excisable goods [124].

Some preliminary findings suggest that ethanol that was officially produced for the production of medicinal compounds was not only illegally sold as surrogate alcohol by single individuals, but also diverted for the production of cheap counterfeit beverages by underground businesses [23,68,69]. The “elimination of the illegal market” as one of the main targets of the concept clearly remains as “not achieved” as counterfeit alcohol purchases were still common in Russia in the past years, together with the misuse of surrogate alcohols as demonstrated by the Irkutsk mass methanol poisonings and the remaining availability of surrogate alcohol in its aftermath [31,64,66,68,124]. Still, the introduction of a de facto minimum price on non-beverage alcoholic products, namely the 2018 decree that prohibits selling them below the established minimum price for vodka, seems to have decreased their availability [64].

Better surveillance and monitoring procedures for unrecorded alcohol are needed, although much has been achieved in this regard in recent years. For instance, in 2019, the Ministry of Health issued official guidelines for assessing total alcohol consumption among adults (18 years and older), which included official sales data as reported by the Federal Service for Alcohol Market Regulation as well as estimates of unrecorded alcohol consumption for all Russian regions for 2017 [76,125]. The large variation of the estimated levels of unrecorded consumption in different regions as well as the existing national framework of measures addressing unrecorded alcohol implies that its regional availability is stemming not from the lack of the specific regulations but the lack of their enforcement.

The evaluation has revealed not only the need for improved and clear referral mechanisms between PHC and narcology services, but also the need for broader frameworks for the monitoring of the harmful use of alcohol, which is a priority task of the concept. As has been outlined, routine screening procedures for risky alcohol use as well as monitoring thereof have been implemented at the level of PHC facilities, but for now, they remain as part of dispanserization protocols only. Although it is recommended that some professional groups—and in some cases, it is even mandatory—regularly undergo dispanserization procedures, it is important to acknowledge that dispanserization does not cover all adult individuals in Russia and that the data analyzed here might be somewhat biased towards individuals of older age, who are monitored because of already diagnosed chronic noncommunicable diseases such as diabetes or cardiovascular diseases [120]. The same can be said for state-run narcological services that monitor only certain population groups. Various studies on narcological patients highlight that individuals with alcohol use disorders do not want to reach out to specialized narcological help for as long as possible to avoid official registration procedures and its limitations and prefer out-of-pocket payments for private narcology services as long as they can afford it. Therefore, a large proportion of narcological patients represent individuals from lower socioeconomic strata with severe forms of AUDs and various comorbidities, who are at higher risk of harm because of social deprivation and AUD chronification [23,68,126]. This means that both dispanserization and narcological treatment data do not seem to adequately account for individuals who are at a certain level of risk because of their drinking but do not meet the diagnostic criteria of AUD.

Therefore, introducing screening and brief interventions for alcohol use at PHC level beyond dispanserization as well as within other settings seems to be an important step towards reducing harmful alcohol use at the population level. It should be explored whether short routine screenings could be introduced not only as part upon first contact with general practitioners, but also in specialized units such as trauma care, cardiology, etc. As mentioned earlier, the AUDIT has been just recently validated for use in Russian PHC facilities and, between it and its short 3-item version, a reliable alcohol screening instrument is now available to better initiate advice, brief or other interventions, and referral to narcological care [104,105]. The developments will contribute to further improvements in this area and create the needed mechanisms and processes to ensure that people who drink at risky or harmful levels can be adequately supported in the health system in order to change their behavior.

Improving the cooperation between specialized and PHC services and establishing specific referral and re-referral protocols based on risk levels seems to be a necessary step, which, however, should go in parallel with changing narcological monitoring provisions to avoid stigmatization.

## 5. Conclusions

With almost half of all deaths in working-age men being attributable to hazardous drinking, Russia was facing an enormous burden of ill health and lost productivity due to alcohol use at the beginning of the 2000s [7]. However, the country was able to turn the tide in the mid-2000s, mainly due to the specific measures aimed at reducing unrecorded alcohol use and reducing its affordability, although a nuanced analysis on the effect of distinct measures has not yet been carried out and seems unfeasible given the available data, as various measures were implemented at the same time [11,21,30,32,33]. The 2009 concept seems to mark an important turning point in Russian alcohol control as it provides a clear public health-oriented and evidence-based framework for its implementation, which is, for its largest part, aligned with the WHO policy portfolio.

In terms of reducing both alcohol consumption and alcohol-related harms, Russia might be one of the brightest examples in the WHO European Region, but it is among many other Eastern European and Central Asian countries which have adopted stricter alcohol control measures as part of legislative frameworks to incorporate health policies and into law, aimed at changing health lifestyles and behaviors and to increase healthy life expectancy [127,128,129,130]. For instance, for the period 2012–2020, all the Member States of the Commonwealth of Independent States have adopted specific strategies to reduce noncommunicable diseases and the associated risk factors at the population level, putting the notion of a healthy lifestyle at the forefront. Together with specific reforms and reorganization processes of the local healthcare systems, especially in relation to AUD treatment and risk prevention, this seems to mark a distinctive departure from the Soviet-style paternalistic approaches to health choices and behaviors.

It remains to be seen what the next Russian alcohol strategy will bring, as the current concept expires this year. But, based on the evaluation’s results, following the same clear structure of aligning the components with the WHO policy portfolio and specifying priority tasks as well as specific indicators to be achieved, as with the 2009 concept, is recommended.

Although all of the bust buys have been implemented in Russia, more can be done in the respective areas: (1) further increase excise tax rates and minimum prices and introduce automatic excise rate adjustment to inflation to ensure that alcohol affordability stays low over time, (2) further restrict marketing of alcoholic beverages, especially in the form of hidden techniques and at the points of sales, develop new enforcement mechanisms of digital marketing restrictions using new technologies, prohibit alcohol sponsorship of sports events, and (3) further restrict availability of retailed alcohol through introducing restrictions on density of outlets (especially in residential areas), introducing restrictions on days of sale, consider increasing the minimum age of drinking, and further decrease hours of sales, especially in regions where alcohol-attributable harms are known to be higher than the national average.

## Figures and Tables

**Figure 1 ijerph-17-08270-f001:**
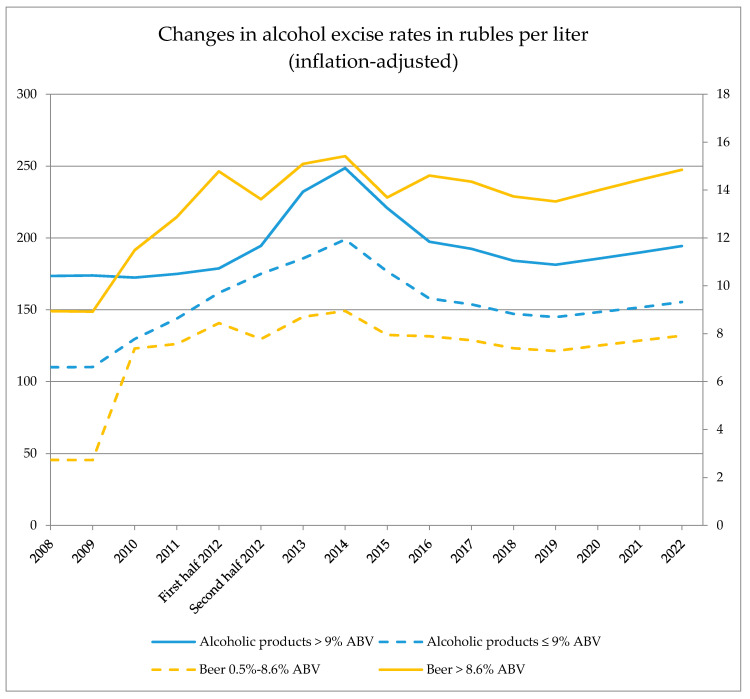
Changes in excise rates on different alcoholic beverages in rubles per liter, adjusted for inflation by using the annual consumer price index. Right scale: beer, left scale: other alcoholic products. Separate excise rates for wine and sparkling wine apply. See Appendix A
Table A4a,b for more details and an overview of non-adjusted excise rates. Sources: Tax Code of the Russian Federation and the Federal Statistical Service.

**Figure 2 ijerph-17-08270-f002:**
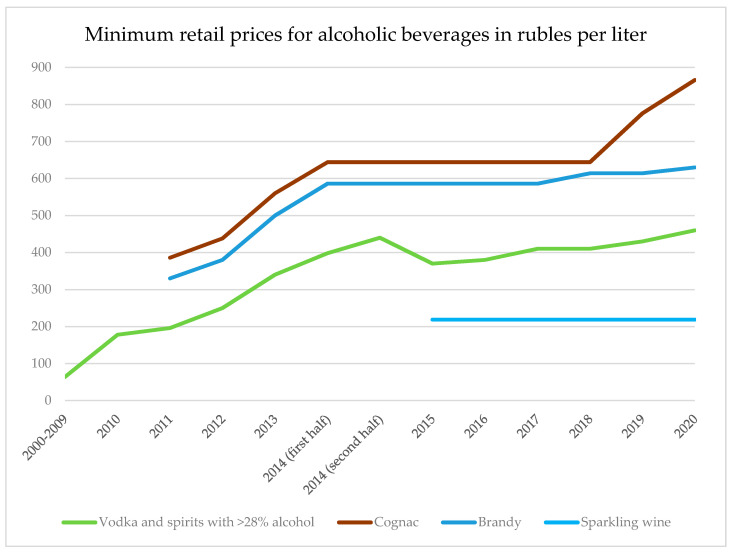
Minimum retail prices for different alcoholic beverages in rubles per liter. Source: Ministry of Finance of the Russian Federation.

**Figure 3 ijerph-17-08270-f003:**
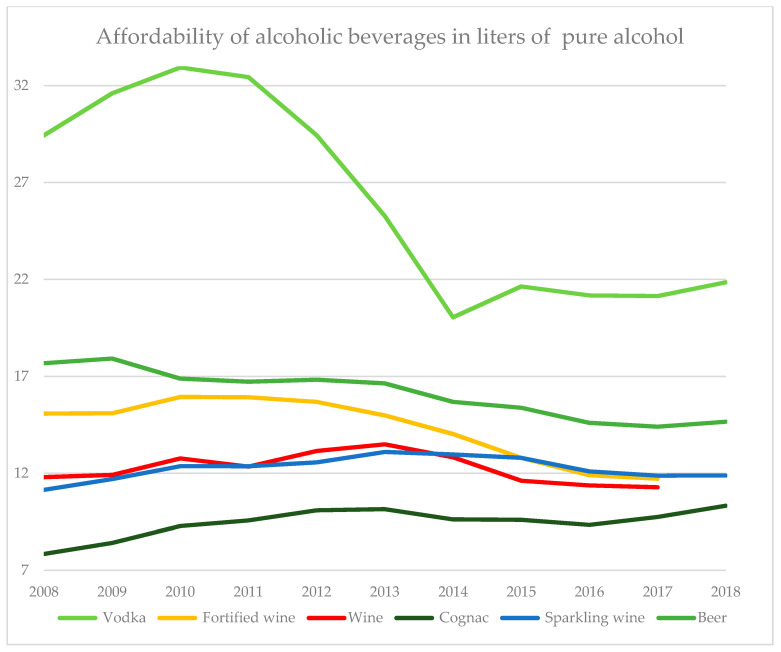
Affordability of alcoholic beverages, defined as how much of a specific beverage could be afforded with a Russian per capita income in a given year, calculated for pure liters of alcohol of each beverage. For wine and fortified wine, no data was available for 2018. Sources: Federal State Statistics Service.

**Figure 4 ijerph-17-08270-f004:**
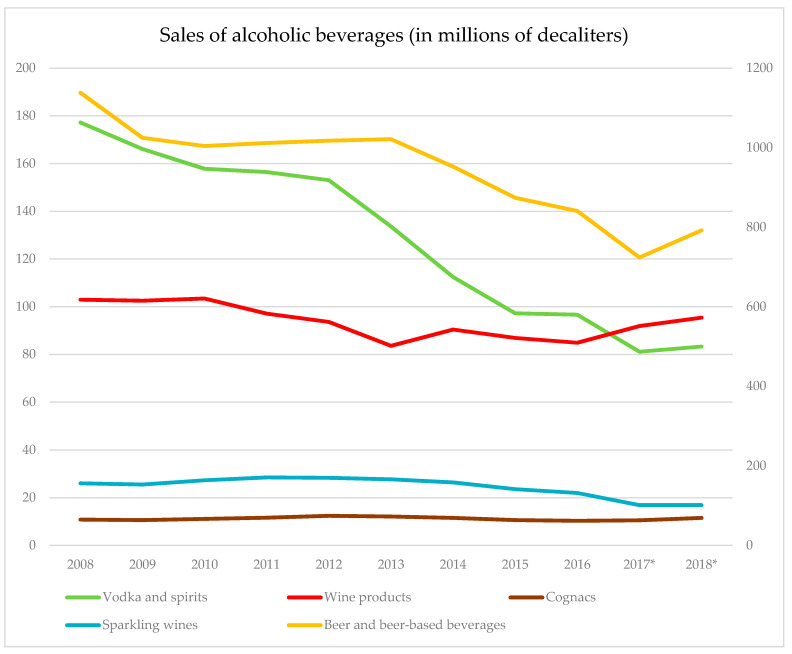
Sales of alcoholic beverages in millions of deciliters. Left scale: vodka and spirits, wine and wine products, sparkling wines. Right scale: beer and beer-based products. Sources: Federal State Statistics Services, beginning in 2017, Federal Service for Alcohol Market Regulation as based on the EGAIS* monitoring.

**Figure 5 ijerph-17-08270-f005:**
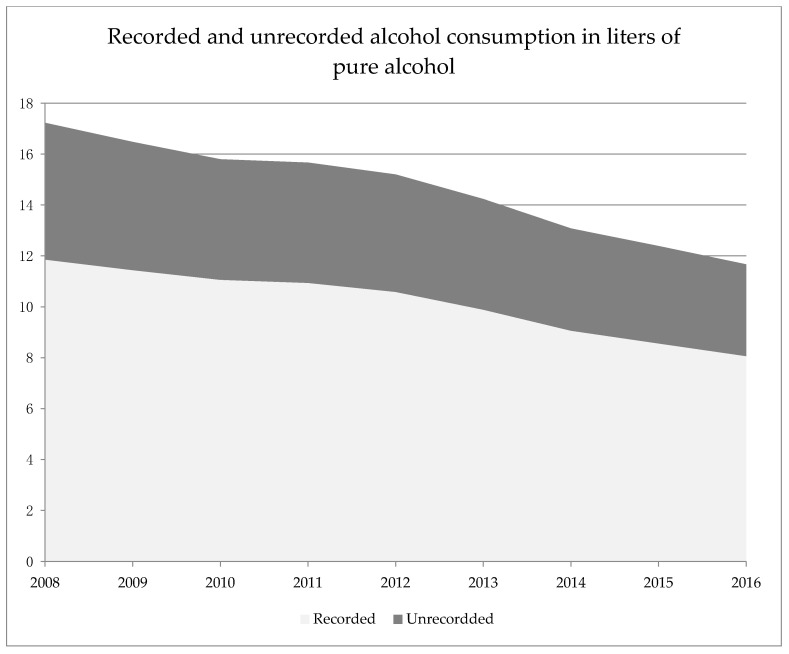
Shares of recorded and unrecorded alcohol consumption among total consumption (15+). Adapted from: World Health Organization Regional Office for Europe, 2019 [11], and Manthey et al., 2019 [77].

**Figure 6 ijerph-17-08270-f006:**
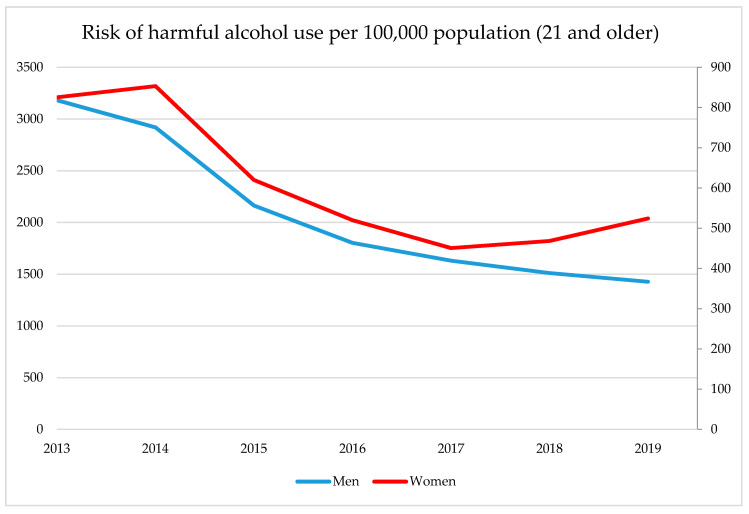
Age-standardized rate of harmful alcohol consumption per 100,000 population (aged 21 and older), defined as a score of ≥2 according to the CAGE screening tool (until 2018) and as ≥3 for women and ≥4 for men for AUDIT-C (starting from 2018). Left scale: men, right scale: women. Source: National Medical Research Center for Therapy and Preventive Medicine of the Ministry of Health of the Russian Federation.

**Figure 7 ijerph-17-08270-f007:**
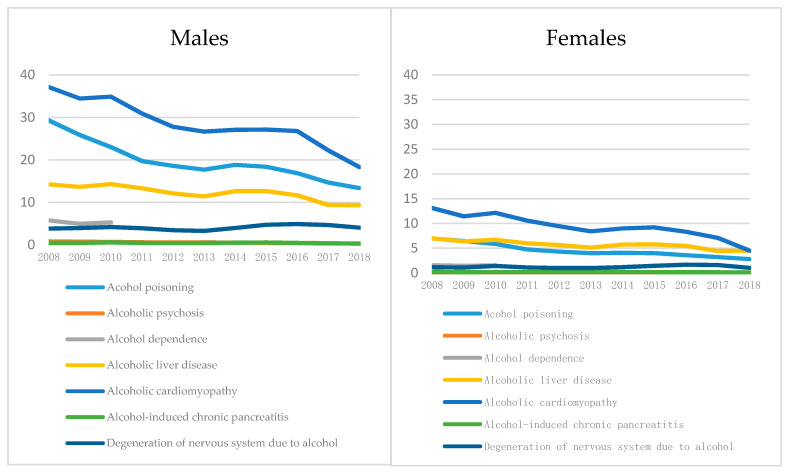
Age-standardized death rates per 100,000 for 100% alcohol-attributable causes of death. Source: Russian Fertility and Mortality Database. Adapted from: World Health Organization Regional Office for Europe, 2019 [11].

**Table 1 ijerph-17-08270-t001:** Overview of the measures of the concept and rating of their implementation (process evaluation).

Measures of the Concept	Related Area of Policy Action of the WHO Portfolio	Component Rated as Evaluable within the Framework	Methods of Evaluation and Data Sources	Measures Rated as Implemented
Media campaigns strengthening public support to combat alcohol abuse.	Leadership, awareness, and commitment	Partly	Media analysis.	Yes
Coordinating training programs and campaigns to promote a healthy lifestyle.	Community and workplace action	Partly	Document analysis of respective documents and provisions.	Yes
Promotion of a healthy lifestyle among children and youth.	Leadership, awareness, and commitment	Partly	Media analysis. Document analysis of respective documents and provisions.	Yes
Alcohol-awareness campaigns among youth.	Leadership, awareness, and commitment	Partly	Media analysis. Document analysis of provisions and existing curricula.	Yes
Prevention of alcohol consumption as part of educational programs and curricula.	Community and workplace action	Partly	Document analysis of provisions and existing training materials.	Yes
Facilitating access to family-friendly treatment and prevention programs.	Health services’ response	Partly	Document analysis of healthcare provisions and the regulatory base, Central Public Health Research Institute of the Ministry of Health of the Russian Federation.	Partly
Improving specialized narcological care for individuals with alcohol use disorders.	Health services’ response	Partly	Quantitative analysis of treatment data, Serbsky National Medical Research Centre for Psychiatry and Narcology of the Ministry of Health of the Russian Federation.	Yes
Creation of rehabilitation and psychological centers for the prevention of alcohol use disorders, capacity building of specialized psychologists.	Health services’ response	Partly	Document analysis of provisions and existing curricula, Serbsky National Medical Research Centre for Psychiatry and Narcology of the Ministry of Health of the Russian Federation.	NA/No information found
Work-place based education, prevention, early detection, and treatment of alcohol use disorders.	Community and workplace action	Partly	Document analysis, Central Public Health Research Institute of the Ministry of Health of the Russian Federation.	Yes
Implementation of pricing policies, based on alcohol content in products	Pricing polices	Yes	Affordability analysis, Federal State Statistics Service, Tax Code of Russian Federation.	Yes
Technical regulations on safety requirements of alcoholic beverages and alcohol-containing products.	Not aligned with the WHO portfolio; partly related to the area of unrecorded alcohol	Yes	Document analysis, Federal Agency on Technical Regulating and Metrology, Eurasian Economic Commission.	Partly (when accounting for regulation of unrecorded)
Restricting hours and places of retail sale of alcohol	Availability of alcohol	Yes	Document analysis, Federal Service for Alcohol Market Regulation.	Yes
Restricting hidden advertising of alcoholic beverages	Marketing of alcoholic beverages	Yes	Document analysis, Federal Service for Alcohol Market Regulation.	Yes
Content-related restrictions of alcohol marketing	Marketing of alcoholic beverages	Yes	Document analysis, Federal Service for Alcohol Market Regulation.	Yes
Restricting events promoting the consumption of alcoholic beverages.	Marketing of alcoholic beverages	Yes	Document analysis, Federal Service for Alcohol Market Regulation.	Yes
Supporting production of high-quality domestic wines	Not supported by existing evidence, not aligned with the WHO portfolio	Partly	Document analysis, Federal Service for Alcohol Market Regulation, Federal Service for Alcohol Market Regulation.	NA
Strengthening administrative responsibility for violations in production and sale of alcohol, including minimum legal age provisions	Reduction of the public health impact of illicit alcohol and informally produced alcohol; Availability of alcohol;	Yes	Document analysis, Federal Service for Alcohol Market Regulation.	Yes
Measures against illegally produced alcohol, strengthening state control over the production and sale of alcohol.	Reduction of the public health impact of illicit alcohol and informally produced alcohol;	Yes	Document analysis, Federal Service for Alcohol Market Regulation.	Yes
Supporting public and religious organizations’ initiatives against alcohol abuse.	Community and workplace action	Partly	Document analysis, Official website of the Russian Government.	Yes
Improving monitoring systems and evaluating the effectiveness of policy implementation to reduce alcohol abuse.	Monitoring and surveillance	Yes	Document analysis, Federal Service for Alcohol Market Regulation, Central Public Health Research Institute of the Ministry of Health of the Russian Federation.	Yes
Development of regional pilot projects to facilitate the implementation of the current concept.	Community and workplace action	Partly	Document analysis, Federal Service for Alcohol Market Regulation, Central Public Health Research Institute of the Ministry of Health of the Russian Federation.	Yes

**Table 2 ijerph-17-08270-t002:** Overview of the priority tasks of the concept, their targets, and rated rate of achievement (effect evaluation).

Priority Task of the Concept and Their Targets	Component Rated as Evaluable within the Framework	Methods of Evaluation and Data Sources	Task Rated as Achieved and the Main Results
Establishing a system to monitor harmful use of alcohol	Yes	Document analysis of provisions and the regulatory base and quantitative analysis, National Medical Research Center for Therapy and Preventive Medicine of the Ministry of Health of the Russian Federation.	Partly. Introduction of the CAGE Substance Abuse Screening Tool in protocols of preventive healthcare in 2013, introduction of the AUDIT since 2018, Russian validation of the AUDIT in 2019. Screening is currently part of dispanserization only.
A relative reduction of spirits consumption with a reduction of total alcohol per capita consumption	Yes	Document analysis of provisions and the regulatory base, Federal Service for Alcohol Market Regulation, Federal State Statistics Service, World Health Organization.	Yes. For the period 2008–2016, total alcohol consumption decreased by 32% according to WHO. According to national data, sales of recorded alcohol declined by 34% for 2008–2018. Relative share of vodka in all sales declined from 47% to 36%.
Promotion of a sober and healthy lifestyle	Yes	Document analysis of provisions and the regulatory base, Ministry of Health, Government of the Russian Federation, Administration of the President of Russia.	Yes. Adoption of the Priority Project “Formation of a healthy lifestyle” and the Federal Project “Formation of a system of motivating citizens to a healthy lifestyle, including healthy eating and giving up bad habits” Adoption of the “Strategy for healthy lifestyle, prevention and control of non-communicable diseases for the period up to 2025”. Reinstating the all-Russian “Sobriety Day”.
Formation of public disapproval of alcohol abuse	Yes	Analysis of sociological surveys on polls on drinking behaviors and attitudes towards alcohol, Russian Public Opinion Research Center, World Health Organization.	Yes. According to WHO, an increase in abstainer rates for the population (15+). For the period 2008–2016 from 23% to 39% (with a 10% relative increase in lifetime abstainers) in men and from 33% to 45% (with a 12% relative increase in lifetime abstainers) in women. Abstainer rates increase as per public opinion polls and national surveys.
Programs for alcohol use disorders prevention	Partly	Document analysis of provisions and existing curricula, Serbsky National Medical Research Centre for Psychiatry and Narcology of the Ministry of Health of the Russian Federation.	Partly. Published materials from narcologists exist, but not possible to evaluate the ongoing training programs. No routine training carried out for PHC health professionals at the moment.
Public awareness on the negative consequences of alcohol abuse	Yes	Analysis of sociological surveys on polls on drinking behaviors and attitudes towards alcohol, Russian Public Opinion Research Center (WCIOM), data on drinking status from the World Health Organization.	Yes. Various awareness campaigns are carried out. Awareness on the harm of all types of alcoholic beverages is growing as documented by public opinion polls.
Incentives for public initiatives aimed at improving public health	Yes	Document and media analysis of NGOs.	Yes The set-up program of the Presidential Grants Fund provides incentives for public initiatives aimed at promoting a healthy lifestyle.
Improving narcological care for individuals with alcohol use disorders	Yes	Document analysis of existing legal documents on narcological care provisions.	Partly. Informed consent was introduced as the basis for treatment and some of the narcological monitoring requirements were amended. Materials that address drinkers at risk were developed by narcologists, but the system remains highly specialized and separated from other health settings. The proportion of individuals receiving alcohol use disorders treatment from private providers and the associated quality of treatment remains unknown.
Promoting physical activity, tourism, and a healthy lifestyle; among children and youth	Yes	Results of the Health Behavior in School-Aged Children Study on physical activity and alcohol consumption and national data. Media and content analysis of websites if the Ministry of Health and Ministry of Sports and Tourism and the Federal Agency for Youth Affairs.	Yes. Various programs launched to Involve youth into playing sports and living a healthy lifestyle, conducting activities in the field of preventing various forms of addictive behaviors. Physical activity time increased in school curricula.
Organization of recreational and leisure activities not related to drinking	Partly	Document analysis of regulations and provisions on city structure	Partly. Various programs launched that aim at increasing physical activity in cities, such as developing a bicycle infrastructure, outdoor gyms, organizing marathons, etc. The WHO Healthy Cities Project is launched in Russia. However, activities are mainly limited to the larger cities for now.
Increasing employment and motivation to work, providing opportunities for cultural leisure activities for rural populations	Partly	Document analysis of regulations, Federal State Statistics Service on overall employment rates.	Partly. A system of Presidential Grants created to incentivize small-scale business ideas and innovation. Special focus is put on projects from rural areas that focus on the installment of sports facilities and promote physical activity. A more detailed analysis of the according enforcement and project realization was not possible. Employment rates slightly increased during the period of interest.
Combating illegal production and sale of alcoholic products, increasing the efficiency of alcohol market regulations	Partly	Document analysis of provisions and the regulatory base, Federal Service for Alcohol Market Regulation, Federal State Statistics Service, and World Health Organization.	Partly. Establishment of the EGAIS system and imposing harsher penalties for illegal production and sales as part of changes to the Administrative and Criminal Codes could reduce unrecorded alcohol consumption, but its share among total alcohol use remains stable as some gaps in the framework exist.
Pricing and taxation measures to reduce affordability of alcoholic beverages, especially for young people	Yes	Document analysis of provisions and the regulatory base, Federal Service for Alcohol Market Regulation, Federal State Statistics Service, Tax Code of the Russian Federation	Yes. Affordability declined over time, although fluctuations were observed due to inconsequent increases in taxation and minimum prices. In recent years, affordability of spirits was increasing again, and tighter price regulation is needed.
Corporate responsibility for producers of alcoholic beverages	Partly	No indicators available that could assess the improvement in effectiveness besides changes in legislation addressing unrecorded alcohol.	Partly. Criminal liability introduced for producers of illegal alcohol, especially if their products have led to serious health damage or death of a consumer.

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
