# Peer review of "Russia’s National Concept to Reduce Alcohol Abuse and Alcohol-Dependence in the Population 2010–2020: Which Policy Targets Have Been Achieved?"

_ijerph, 2020, doi:10.3390/ijerph17218270_

Round 1

Reviewer 1 Report

Excellent study. Well-written manuscript. Very important results. I just have a few comments and suggestions.

(1) I suggest that you insert in the Abstract some of the numerical declines in alcohol measures that you show in the Appendix: increase in minimum price of alcohol; decrease in alcohol consumption per capita; decrease in harmful use of alcohol.

(2) Concerning the data mentioned above and other data in the Appendix, were you able to verify the reliability of the data?

(3) Please fix a few typos: line 130 should be "..and the key indicators"; line 204 should be "...amendments were made.." line 263 should be "..and temporarily loosened.." line 279 shouldn't this be "...higher for higher alcohol content beverages..?; line 283 what does "as week as.." mean? Should it be "as well as"?  line 511 should be "..within one year or if the patient.." line 564 no space in "changes" line 741 should be "..best buys"

(4) The figures you show are great. I wish some in the Appendix could be in the main text.   

(5) Appendix A: Were you able to obtain any measures of enforcement of these law and policy changes?

Author Response

Reviewer(s)' Comments to Author:

Reviewer 1

Excellent study. Well-written manuscript. Very important results. I just have a few comments and suggestions.

  1. (1) I suggest that you insert in the Abstract some of the numerical declines in alcohol measures that you show in the Appendix: increase in minimum price of alcohol; decrease in alcohol consumption per capita; decrease in harmful use of alcohol.

A1: We have included some numerical information in the abstract, specifically on the rated tasks and the decreases in alcohol per capita consumption by about a third for the period of the last ten years. We also added the proportional decline of sales in the manuscript text (ll.351-352).
However, we are hesitant to include the numerical information on the minimum pricing because, for international audiences, this information—taken out of context and without the accompanying affordability graph—might signal significant increases, while in reality, and accounting for inflation, they are not as impressive as they might appear in the abstract. We also have similar reservations about the decreases in the harmful use of alcohol because screening instruments were changed and because we have overall doubts about the quality of data, which were described in greater detail in the limitation section (ll.646-648).

  1. (2) Concerning the data mentioned above and other data in the Appendix, were you able to verify the reliability of the data?

A2: We have included only the data that we considered to be reliable overall (for instance, data taken from the Federal State Statistical Service of the Russian Federation as a credible source) and we have clearly stated in the limitation section what the specific concerns on data reliability were. We made the specific concerns of the “risk of harmful use of alcohol” indicator clearer in the limitations section. Despite these concerns, we believe that it is of great value to report this indicator as it adds to the ‘bigger picture.’

  1. (3) Please fix a few typos: line 130 should be "..and the key indicators"; line 204 should be "...amendments were made.." line 263 should be "..and temporarily loosened.." line 279 shouldn't this be "...higher for higher alcohol content beverages..?; line 283 what does "as week as.." mean? Should it be "as well as"?  line 511 should be "..within one year or if the patient.." line 564 no space in "changes" line 741 should be "..best buys"

A3: The typos have all been fixed. The increase in taxation is indeed stronger/higher for lighter alcoholic beverages, but because this seems to not be apparent from the figure, we have added numerical information on this (ll. 279-284).

  1. (4) The figures you show are great. I wish some in the Appendix could be in the main text.   

A4:We have moved the figures on unrecorded alcohol use and mortality from 100% alcohol-attributable causes of deaths to the main text.

  1. (5) Appendix A: Were you able to obtain any measures of enforcement of these law and policy changes?

A5: Enforcement is an important dimension of alcohol policy and it is much more difficult to measure than legislative frameworks and implementation data and, unfortunately, there aren’t enough studies and data available on this topic for Russia. We have tried to reflect this in the section on unrecorded alcohol, drawing on our own experience in the field, but we decided not to delve too deeply into this area in this manuscript.

We thank the reviewers for their comments and hope that we were able to address all of them.

For the authors,

Maria Neufeld

Reviewer 2 Report

This article is very well written and provides a comprehensive review of policy/campaign related to alcohol. I rarely recommend a manuscript to be published as it is but this one is pretty close. If I have to comment, I will point out that the association between pricing policy and sales is not analyzed in a statistical model. The authors seem to have data available to do so. However, a comprehensive review in a topic that we know little of by itself is merit enough to warrant publication. Good job!

Author Response

Reviewer 2

  1. (1) This article is very well written and provides a comprehensive review of policy/campaign related to alcohol. I rarely recommend a manuscript to be published as it is but this one is pretty close. If I have to comment, I will point out that the association between pricing policy and sales is not analyzed in a statistical model. The authors seem to have data available to do so. However, a comprehensive review in a topic that we know little of by itself is merit enough to warrant publication. Good job!

A1: This was mainly a descriptive study and we did not perform the mentioned analysis as part of this contribution as this was outside of the announced scope. However, we will consider this for future publications.

We thank the reviewers for their comments and hope that we were able to address all of them.

For the authors,

Maria Neufeld

Reviewer 3 Report

The paper deals with the actions taken by Russian’s Government in order to mitigate the social effects of the alcohol consumption during last decades.

The manuscript is well written and sounds scientifically relevant. The bibliography discussion is exhaustive and the results are presented and discussed in detail.

I didn’t encounter any particular issue against the publication in present form, just a general observation. Russia territory is wide and different from part to part. Have you considered the possibility to include a geographical effect within the parameters related with alcohol consumption? Can alcohol consumption be affected by the climate and the social situation of isolated part of Russia?

I suggest adding some short comments in that sense in order to complete the manuscript.

Author Response

Reviewer 3

  1. (1) The paper deals with the actions taken by Russian’s Government in order to mitigate the social effects of the alcohol consumption during last decades.

The manuscript is well written and sounds scientifically relevant. The bibliography discussion is exhaustive and the results are presented and discussed in detail.

I didn’t encounter any particular issue against the publication in present form, just a general observation. Russia territory is wide and different from part to part. Have you considered the possibility to include a geographical effect within the parameters related with alcohol consumption? Can alcohol consumption be affected by the climate and the social situation of isolated part of Russia?

I suggest adding some short comments in that sense in order to complete the manuscript.

A1: Overall, we agree that there is a huge regional variety in Russia, simply because this is the world’s largest country and a multinational state with more than 180 ethnic groups, and with more than 30 official languages and more than 100 “minority” languages. Also, almost 12% of the population are Muslim and this proportion varies significantly across the different regions. The latter is probably one of the largest factors shaping alcohol use, besides the already discussed regional differences in alcohol sale times. We have added this comment on regional differences to the limitations section (pp.660-666).

As for the geographical differences, we are not aware of such studies, but a recent study on regional differences in alcohol intake in Russia has revealed that “coefficients for the average temperature in January [… ] were not statistically significant.”

Zasimova, L., & Kolosnitsyna, M. (2020). Exploring the relationship between drinking preferences and recorded and unrecorded alcohol consumption in Russian regions in 2010–2016. International Journal of Drug Policy82, 102810.

We thank the reviewers for their comments and hope that we were able to address all of them.

For the authors,

Maria Neufeld